# Impact of Dietary Supplementation of Cysteamine on Egg Taurine Deposition, Egg Quality, Production Performance and Ovary Development in Laying Hens

**DOI:** 10.3390/ani13193013

**Published:** 2023-09-25

**Authors:** Jing Chen, Youli Wang, Zhenhai Tang, Xiaorui Guo, Jianmin Yuan

**Affiliations:** 1Key Laboratory of Qinghai-Tibetan Plateau Animal Genetic Resource Reservation and Utilization, Ministry of Education, Chengdu 610041, China; chenjing96518@163.com; 2Key Laboratory of Sichuan Prpvince for Qinghai-Tibetan Plateau Animal Genetic Resource Reservation and Exploitation, College of Animal Science and Veterinary Medicine, Southwest Minzu University, Chengdu 610041, China; 3Sichuan New Hope Liuhe Technology Innovation Co., Ltd., Chengdu 610100, China; 4State Key Laboratory of Animal Nutrition, College of Animal Science and Technology, China Agricultural University, Beijing 100193, China; embraceuu@163.com (Z.T.); gxrfy0820@163.com (X.G.); yuanjm@cau.edu.cn (J.Y.)

**Keywords:** cysteamine, laying hens, yolk taurine content, egg quality, production performance

## Abstract

**Simple Summary:**

Taurine is a necessary amino acid for human health, while cysteamine is an intermediate metabolite to the synthesis of taurine. This study investigated effects of dietary cysteamine supplementation on the egg taurine deposition efficiency, egg quality, production performance and ovary development in laying hens. The results of this study indicate that cysteamine supplementation benefits yolk taurine deposition in both peak and late stages of egg production, but hens at the late stage of egg production showed depressed production performance and egg quality. The present study revealed that laying hens at the peak stage of egg production are suitable for cysteamine diets to produce high-taurine eggs.

**Abstract:**

This study aimed to examine the effect of dietary cysteamine on yolk taurine content in hens during different egg production periods. In Exp. 1, China Agricultural University-3 (*CAU-3*) hens at the peak stage of egg production (aged 31 wks) were used to explore the effect of diets supplemented with 0.1% cysteamine on yolk taurine content, egg quality and production performance. In Exp.2, two breeds of hens (half Hy-Line Brown and half *CAU-3* hens) at the late stage of egg production (68 wks) were used to investigate the influence of diets supplemented with 0, 0.02%, 0.04%, 0.08% or 0.10% cysteamine on yolk taurine content, egg quality, production performance and ovary development. In Exp.1, diets supplemented with 0.1% cysteamine significantly increased yolk taurine content (*p* < 0.05) without negative influence on production performance or egg quality. In Exp.2, the highest yolk taurine content was observed when cysteamine was supplemented at 0.08% (*p* < 0.001). However, supplemental cysteamine linearly or quadratically decreased production performance over the first few weeks of feeding, and the effects disappeared with continued feeding (*p* < 0.05). In conclusion, this study indicated that cysteamine supplementation benefits yolk taurine deposition in hens at both peak and late stage of egg production, but hens at the late stage of egg production show depressed production performance and egg quality.

## 1. Introduction

Taurine, a free amino acid that widely exists in animals, is necessary for normal body functioning [1]. It is well established that deficits of taurine are associated with multiple diseases such as myocardiopathy, renal dysfunction, dysplasia and retinal neuron injury [2,3]. Additionally, studies have shown that taurine supplementation protects against mitochondrial defect pathologies, such as neurological disorders [3], cancer [4], cardiovascular diseases [5], metabolic syndrome [6] and aging [7]. Therefore, producing taurine-enriched foods may not only provide fundamental dietary nutrients but may also help humans in improving taurine intake and may be beneficial for human health.

Even though it has been shown that diets supplemented with taurine can improve the taurine deposition in egg yolks [8] and can have benefits for kidneys [9], the cost of dietary taurine supplementation is significant. So, other methods have been used to improve taurine deposition in animal production. Studies have shown that diets supplemented with feather meal can improve the taurine content in pork [10] and eggs [11]. Additionally, it has been shown that diets supplemented with methionine can increase serum taurine content along with increased egg weight and egg mass [12]. Moreover, Wei et al. observed that lactating sows fed diets with increasing methionine could improve plasma and milk taurine content [13]. These results indicate that sulfur-containing amino acid supplementation may promote taurine deposition. However, feather meal is susceptible to salmonella contamination that may pose a danger to animals [14]. Alternatively, it is well known that taurine can be synthesized by the following two oxidation pathways: (1) the conversion of cysteine to cysteine sulfinate by cysteine dioxygenase, followed by its decarboxylation to hypotaurine by cysteine sulfinic acid decarboxylase and the oxidation of hypotaurine to taurine; and (2) the incorporation of cysteine into CoA, followed by the release of cysteamine during CoA turn over, the oxidation of cysteamine to hypotaurine by aminoethanethiol dioxygenase and the further oxidation of hypotaurine to taurine [15,16]. Cysteamine is an intermediate metabolite to the synthesis of taurine in the second pathway; therefore, diets directly supplemented with cysteamine may reduce the metabolic steps for taurine synthesis. However, it is not known if dietary cysteamine supplementation can improve the taurine deposition efficiency in eggs. 

As eggs are popular among the public and are not subject to the dietary exclusion of religious observers, we aimed to study the impact of cysteamine supplementation on egg taurine deposition in hens at different stages of egg production. In the present study, two experiments were conducted. The Exp.1 was designed to explore the impact of cysteamine on laying performance, egg quality and egg taurine content in China Agricultural University-3 (*CAU-3*) hens at the peak stage of egg production, and in Exp.2, Hy-Line Brown hens and *CAU-3* hens at the late stage of egg production were used to explore proper dietary cysteamine dose and different hen breeds for a better taurine deposition. 

## 2. Materials and Methods

All animal procedures were conducted according to the guiding principles of the Animals Care and Ethics Committee of Southwest Minzu University (Approval code: SMU-CAVS-200603011).

### 2.1. Birds, Feeding, Sampling Systems 

In Exp.1, one hundred and eighty *CAU-3* laying hens at the peak stage of egg production (aged 31 wks) with similar body weights were randomly assigned to 2 diet treatments: a basal diet and a basal diet with 0.1% cysteamine supplementation (cysteamine was supplied as coated cysteamine hydrochloride by Hangzhou King Techina Technology Co., Ltd. (Hangzhou, China), which contained 27% cysteamine hydrochloride). The basal diet for hens at the peak laying period is shown in Table 1. Each treatment had 6 replicates (each replicate had 5 cages with 3 hens in each cage). Ladder cages held four chickens each (45 cm × 45 cm × 45 cm). At the same time, to ensure that the chicken coop was closed and ventilated, the average relative humidity was routinely maintained at ~55%, and it was ensured that the hens received 16 h of light every day. Feed was provided at 7:00 am and 1:00 pm, and water was provided ad libitum. The feeding amount each day was appropriately adjusted according to the amount of feed remaining from the previous day. This was to ensure no feed was leftover in the trough each night and prevent picky eating. The experiment lasted for 5 weeks. The egg numbers and egg mass of each pen were recorded daily, feed intake was recorded weekly and laying performance was calculated weekly. Eggs at d 8, 10, 14 and 21 were collected, and the Haugh unit, albumen height, yolk color and eggshell strength were measured by using the Egg Multi Tester (Model EMT-7300, Robotmation Co., Ltd., Tokyo, Japan) within 24 h of collection. Based on the formation of new yolks requiring 7 days, egg yolks at d 8 were collected and 3 yolks in each replicate cage were mixed for taurine detection. 

In Exp.2, a total of 180 Hy-Line Brown and 180 *CAU-3* (both aged 68 wk, at the late stage of egg production) hens were mixed and then allocated into 5 dietary treatments (basal diet supplemented with cysteamine at levels of 0, 0.02%, 0.04%, 0.08% or 0.10%) with 6 replicates (each replicate had 2 cages of Hy-Line Brown hens and 2 cages of *CAU-3* hens; each cage had 3 hens). Eggs in each replicate rolled down and then come together. The composition of the basal diet for hens in the late laying period is listed in Table 1. At the same time, to ensure that the chicken coop was closed and ventilated, the average relative humidity was routinely maintained at ~55%, and it was ensured that the hens received 16 h of light every day. Birds were fed ad libitum accompanied with free access to water. The experiment lasted for 8 weeks. The egg numbers and egg mass of each pen were recorded daily, feed intake was recorded weekly and laying performance was calculated every two weeks. As eggshells of Hy-Line Brown hens are brown, and eggshells of *CAU-3* hen are pink, it was easy to distinguish hen breed when collecting and analyzing eggs. This experiment was arranged as a 2 × 5 factorial that was conducted to determine the effects of the 2 breeds and 5 dietary cysteamine levels on egg quality and ovary development. Eggs at wk 4, 5, 6, 7 and 8 of feeding were collected and the Haugh unit, albumen height, yolk color and eggshell strength were measured by using the Egg Multi Tester (Model EMT-7300, Robotmation Co., Ltd., Japan) within 24 h of collection. After 8 weeks of feeding, eggs were collected and 3 yolks in each replicate were mixed for taurine content detection (*n* = 6). At the same time, blood (overnight fasted) was collected from the wing vein then centrifuged at 3000× *g* for serum taurine content detection. Then, one hen from each replicate was selected and euthanized by cervical dislocation. Ovaries were isolated and weighed. The ovary-weight-to-body-weight ratio was calculated as relative ovary weight. Small yellow follicles (SYF, 4 mm ≤ diameter < 8 mm) and large yellow follicles (LYF, diameter ≥ 8 mm) were separated and weighed. Additionally, the numbers of SYF and LYF were recorded. The ratios of SYF and LYF to the ovary were calculated. 

### 2.2. Yolk and Serum Taurine Determination

High-performance liquid chromatography (HPLC, 1260FLD, Agilent, America) was used for taurine detection as follows: yolks were first freeze dried and then 0.25 g yolk or 100 μL serum was sampled and put into 50 mL centrifuge tubes. Then 15 mL ultrapure water was added to mix with the yolk/serum for later ultrasound (20 min). After ultrasound treatment, 200 μL potassium hexacyanoferrate (15%) was added and mixed. Then, 200 μL zinc acetate (30%) was mixed and centrifuged at 10,600× *g* for 10 min. The liquid supernatant was then filtrated with a 0.22 μm hydrophilic filter membrane. The filtrate was then mixed with o-phthalaldehyde (OPA, *v*:*v* = 1:1). An Agilent 1260 system was used to determine taurine levels. An Eclipse plus C18 column (5 μm, 4.6 × 250 mm) was employed. The binary gradient elution system consisted of (A) water and (B) acetonitrile and separation was achieved using the following gradient: 10% B over 0–2 min, 20% B over 2–6 min, 23% B over 6–11 min, 40% B over 11–12 min, 80% B over 12–14 min, 90% B over 14–15 min, and 10% B over 15–20 min. The flow rate was 1 mL/min, and the column temperature was 35 °C. All the samples were kept at 4 °C during the analysis. The injection volume was 10 μL. Retention time at 11.70 min. 

### 2.3. Statistical Analysis

The data were analyzed by SPSS 25.0 (SPSS Inc. Chicago, IL, USA). The data in Exp.1 were analyzed by Student’s *t*-test. In Exp.2, a 2 × 5 factorial arrangement of 2 breeds and 5 dietary cysteamine levels was used, and the main effect of breed, dietary cysteamine level and interactions between breed and dietary cysteamine level were tested by a general linear model. Items with significant interaction effects were further analyzed by ANOVA. Significant means were separated by Duncan’s Multiple Range test method. Labeled means without a common letter are significantly different; *p* < 0.05. Moreover, to investigate the trend between cysteamine and test index in Exp.2, data were pooled from the two breeds of laying hens at the same cysteamine levels, and linear and quadratic responses were analyzed with orthogonal polynomials.

## 3. Results

### 3.1. Impact of Dietary Cysteamine Supplementation on Yolk Taurine Content and Egg Quality in Hens at the Peak Stage of Egg Production

In this study, the diet supplemented with 0.1% cysteamine significantly improved yolk taurine level by 48.68% in *CAU-3* laying hens at the peak stage of egg production (*p* < 0.001, Table 2), relative to the control. Moreover, in laying hens at the peak stage of egg production, diets supplemented with 0.1% cysteamine improved yolk color compared to the control (*p* < 0.01). However, 0.1% cysteamine supplementation did not influence the breaking force, albumen height or Haugh unit (Table 2).

### 3.2. Impact of Dietary Cysteamine Supplementation on Production Performance in Hens at the Peak Stage of Egg Production 

In laying hens at the peak stage of egg production, the diet supplemented with 0.1% cysteamine had no influence on laying rate, egg mass, egg weight or feed conversion ratio during wk 1, 2, 3, 4 or 5, but feed intake was increased for wk 5 (*p* < 0.01, Table 3). 

### 3.3. Impact of Dietary Cysteamine Supplementation on Taurine Content in Yolk and Serum at the Late Stage of Egg Production 

No interaction effect was found between the levels of yolk taurine and serum taurine between breed and cysteamine level (Table 4). Hy-Line Brown hens had greater serum taurine content than *CAU-3* hens (by 24.4%, *p* = 0.005), but no difference in yolk taurine level was found between *CAU-3* hens and Hy-Line Brown hens. A linear increase was found for yolk taurine content with increased cysteamine levels (*p* < 0.001). Moreover, a quadratic increase tendency was observed for both serum and yolk taurine content with increased cysteamine levels (0.1 < *p* < 0.05). Taurine content was highest in both serum and yolk at the 0.08% inclusion of cysteamine in the diet.

### 3.4. Impact of Dietary Cysteamine Supplementation on Production Performance in Hens at the Late Stage of Egg Production

In Table 5, compared to the control, feed intake linearly and quadratically decreased with increased cysteamine supplementation during wk 2 (*p* < 0.001), and a linear decrease was observed for wk 4 (*p* < 0.01), but the effects did not hold for wk 6 and 8. Linear and quadratic effects were found for feed conversion ratio at wk 4 (*p* < 0.001) and a cysteamine supplementation level of 0.1% significantly increased the feed conversion ratio, but these effects were not found at wk 2, 6 or 8. Linear decreases were observed in laying rate at wk 2, 4 and 6 (*p* < 0.001), while quadratic decreases were observed at wk 2 and 4 (*p* < 0.001). Interestingly, at wk 8, we found that cysteamine supplementation level of 0.04 and 0.10% decreased the laying rate, while only the cysteamine supplementation level of 0.08% increased the laying rate. Similarly, egg mass linearly and quadratically decreased with increased dietary cysteamine supplementation at wk 2 and 4 (all *p* < 0.001); however, as feeding continued, the negative effect subsided at wk 6. Moreover, a linear and quadratic increase was observed for egg mass with increased cysteamine supplementation at wk 8 (*p* < 0.001), and the highest egg mass was found with cysteamine supplementation level of 0.08%. Linear and quadratic increases were observed at wk 2, 4 and 6 with increased cysteamine supplementation (all *p* < 0.001), and the cysteamine level of 0.1% produced the highest egg weight. Both the cysteamine supplementation levels of 0.08% and 0.1% showed no negative effect for egg weight, whereas a cysteamine supplementation level of 0.02% or 0.04% decreased the egg weight at wk 2. These results suggest that the negative effects of cysteamine on production performance are mainly observed in the first few weeks of feeding, and the negative effects gradually decrease and positive impacts appear as the feeding continues.

### 3.5. Impact of Dietary Cysteamine Supplementation on Egg Quality in Hens at the Late Stage of Egg Production 

There were interaction effects between breed and cysteamine level in albumen height, Haugh unit and egg yolk color (*p* < 0.01), where albumen height and Haugh unit gradually decreased with increased dietary cysteamine levels (*p* < 0.001, Table 6). However, increased values for yolk color and breaking force were observed in Hy-Line Brown hens compared to *CAU-3* hens (*p* < 0.001, Table 6). Moreover, breaking force gradually decreased with increasing cysteamine supplementation (*p* = 0.031, Table 6). These results suggest that cysteamine supplementation improves the yolk color but decreases the breaking force, albumen height and Haugh unit.

In order to investigate the trend between cysteamine and egg quality, data were pooled from the two breeds of laying hens at the same cysteamine levels, and linear and quadratic responses were analyzed by orthogonal contrasts (Table 6). Results indicated that there were no linear or quadratic relationships between egg quality and increased cysteamine supplementation.

### 3.6. Impact of Dietary Cysteamine Supplementation on Ovary Development in Hen at the Late Stage of Egg Production

In Table 7, interaction effects between breed and cysteamine level were observed for the numbers of SYF and LYF and SYF weight/ovary weight (*p* < 0.05), where the highest numbers of SYF and SYF weight/ovary weight were observed in *CAU-3* hens at a 0.08% cysteamine additive dose, and the highest number of LYF was observed in Hy-Line Brown hens at a 0.1% cysteamine additive dose. Moreover, the relative ovary weight of *CAU-3* hens was larger than that of Hy-Line Brown hens (*p* < 0.001), but no difference was found in LYF/ovary weight. Similar to egg quality, no linear or quadratic response was found for ovary development with increased cysteamine supplementation.

## 4. Discussion

Taurine as a bioactive molecule has been evidenced to be helpful for body health, and functional foods containing taurine are becoming more popular. Cysteamine is an intermediate metabolite to the synthesis of taurine, and diets directly supplemented with cysteamine require reduced metabolic steps for taurine synthesis. This study indicates that cysteamine supplementation at 0.1% improves the yolk color and yolk taurine content without influencing other egg quality and production performance metrics in hens at the peak stage of egg production. Hens at the late stage of egg production show the highest yolk taurine content at 0.08%. However, these late-stage hens show a depressed production performance within the first 6 wk and a depressed egg quality (except yolk color) throughout the experiment (8 wk of feeding experiment).

Taurine can be synthesized by catalysis of cysteine sulfinate decarboxylase. It has been shown that taurine exists in many different tissues in different organisms [1]. In this study, we found that yolk taurine content was enhanced with 0.1% cysteamine supplementation in hens at the peak stage of egg production or with more than 0.04% cysteamine supplementation in hens at the late stage of egg production. This indicates that cysteamine can be used to synthesize taurine in hens at multiple stages of production. Though it has been reported that serum taurine content declines with age in mice, monkeys and humans [7], we found that hens at the late stage of egg production (aged 68 wk) produce more taurine than hens at the peak stage of egg production when fed basic diets. This suggests that hens at the late stage of egg production may have a greater ability to synthesize taurine relative to younger birds. However, with diets supplemented with cysteamine, it seems that the effect induced by the stage of egg production is lost. This may imply that basic diets for hens at the peak stage of egg production are more deficient in raw materials for taurine synthesis than diets for hens at the late stage of egg production. 

In this study, we found that cysteamine supplementation had no significant influence on production performance in hens at the peak stage of egg production. However, in hens at the late stage of egg production, dietary cysteamine showed a negative dose-dependent effect on production performance. It has been reported that supplemental dietary cysteamine at 0.04% increased the laying rate in San Huang broiler breeders at the late stage of egg production [17]. In this study, hens at the late stage of egg production fed a diet with 0.08% cysteamine had an increased laying rate only at wk 8, while other levels of cysteamine supplementation showed depressed laying rates. Conversely, it has reported that a cysteamine supplementation level of 0.04% increased the average laying rate by 2.24% [17]. Our results are similar to a result in Japanese quail, where it was reported that diets supplemented with taurine decreased laying rate [8]. The increased laying rate may be explained by an increased number of SYF during 0.08% cysteamine supplementation, and an increased number of SYF in the cysteamine treatments may be explained by the oocyte meiosis promotion effect of cysteamine [18,19]. However, more supplemental cysteamine caused negative effects on the laying rate and feed conversion ratio, which is similar to the published work [20]. Previous work showed that egg weight was decreased in Single Comb White Leghorn hens fed with diets supplemented with taurine [21], which means taurine may have a negative impact on egg weight. Inconsistently, in this study, egg weight only decreased with cysteamine supplementation levels of 0.02 and 0.04% during wk 2, but it was increased with0.1% cysteamine supplementation. This could be due to natural variations between the different breeds. Cysteamine supplementation at 0.08% or more significantly reduced the feed intake at wk 2, subsequently affecting the laying rate and egg mass. However, this negative effect gradually disappeared as the experiment continued. The reduced feed intake here may be caused by the strong pungent odor of cysteamine resulting in a low palatability [22]. In summary, cysteamine supplementation has no influence on production performance in hens at the peak stage of egg production, while it weakens the production performance during the first few weeks of feeding in hens at the late stage of egg production. It is worth noting that there was no negative effect of cysteamine supplementation on production at the late stage of egg production.

Egg quality includes the index of yolk color, albumen height, Haugh unit, breaking force and so on. Usually, the higher the yolk color measurement of an egg, the more popular it is with consumers, and yolk color is influenced by pigment content in feed, hen breed and antioxidant contents [23]. In this study, cysteamine supplementation elevated the egg yolk color in *CAU-3* hens at the peak stage of egg production. A recent study has shown that some bacteria play a potential role in pigment synthesis [24]. At the same time, it has been reported that cysteamine supplementation changed bacteria composition in the intestine [25] and regulated gut pathogenic bacteria [26]. Additionally, cysteamine supplementation is considered to protect against the oxidation of pigments [27]. So, the improved egg yolk color in hens at the peak stage of egg production may be caused by the cysteamine-induced change in the microbial community and oxidative resistance. Similarly, Hy-Line brown hens had greater values for yolk color at the late stage of egg production. However, cysteamine did not influence the egg yolk color in Hy-Line Brown hens at the late stage of egg production, which is similar to the result observed in San Huang broiler breeders during the post-peak stage of egg production [17]. Albumen height reflects the protein quality and is influenced by storage time and age of hens. Previous study has reported that albumen height is decreased with age and storage time [28]. In this study, cysteamine supplementation decreased albumen height in hens at the late stage of production regardless of hen breed. Consistently, a decreased albumen weight was also reported before in hens at the late stage of egg production (aged 67 wks) when cysteamine supplementation was at 0.04% [17]. A high value of breaking force indicates a hard shell and a reduced likelihood the egg will break. It has been found that cysteamine supplementation decreased the percentage of broken eggs [17], which indicates that cysteamine supplementation increased the breaking force. However, we found that cysteamine supplementation decreased the breaking force in hens at the late stage of egg production. A Haugh unit ≥ 80 indicates a fresh egg, because the Haugh unit decreases as the storage time is extended [29]. In this study, we found that cysteamine supplementation did not influence the Haugh unit in Hy-Line Brown hens at the late stage of egg production, but the Haugh unit decreased in *CAU-3* hens at the late stage of egg production. These results may mean that hen breed can influence the cysteamine metabolism and influence egg quality. Additionally, we found that cysteamine supplementation did not influence the egg quality of hens at the peak stage of the egg production, which means that the effect of cysteamine supplementation on egg quality was also influenced by the stage of egg production.

During the egg-laying period, a single SYF is recruited almost every day from a cohort of SYF to develop into a hierarchal follicle; this process is termed follicle selection. In this study, the number of SYF was more than the number of LYF and the weight of SYF was less than that the weight of LYF, which is consistent with a previous work [30]. In this study, at the late stage of egg production, decreased numbers of LYF were found in *CAU-3* hens when fed a diet supplemented with cysteamine, while decreased numbers of SYF and SYF/ovary weight were found in Hy-Line Brown hens when cysteamine was supplemented at 0.04%. It has been reported that cysteamine has a toxicity to bovine embryos and chicken sperm [31,32]. Additionally, taurine supplementation-induced oxidative stress may also depress ovary development [33]. Therefore, depressed ovary development caused by the toxicity of cysteamine and oxidative stress of taurine may explain the depressed egg quality and production performance in hens at the late stage of egg production. 

## 5. Conclusions 

In this study, we found that dietary cysteamine supplementation at 0.1% improved the yolk taurine content in hens at the peak stage of egg production without influencing egg quality or production performance. However, cysteamine supplementation at 0.08% improved the yolk taurine in hens at the late stage of egg production with depressed production performance and egg quality. 

## Figures and Tables

**Table 1 animals-13-03013-t001:** Diet composition and nutritional value for hens at peak and late laying periods.

Feed Ingredient	Hens at Peak Laying Period	Hens at Late Laying Period
	Content (%)	Content (%)
Corn	56.30	65.70
Soybean meal (Crude protein, 44.2%)	30.70	21.00
Soybean oil	2.00	1.00
Limestone	9.10	10.00
Sodium chloride	0.35	0.30
Dicalcium phosphate	1.00	1.40
Phytase	0.02	0
Antioxidant	0.02	0
Se-yeast (0.1%)	0.04	0
Choline chloride (60%)	0.10	0.06
Mineral premix ^1^	0.20	0.20
DL-Methionine (98%)	0.13	0.22
Vitamins premix ^2^	0.02	0.02
Complex enzyme ^3^	0.02	0
Total	100.00	100.00
Calculated nutrition levels		
Metabolizable energy (Kcal/kg)	2750.00	2700.00
Crude protein (%)	17.50	14.90
Calcium (%)	3.78	4.17
Non-phytate phosphorus (%)	0.27	0.35
Lysine (%)	0.93	0.93
Methionine (%)	0.41	0.43
Methionine + Cysteine (%)	0.70	0.65

^1^ The trace mineral premix provided the following per kg of diets: Cu (CuSO_4_.5H_2_O), 8.00 mg; Zn (ZnSO_4_), 75.00 mg; Fe (FeSO_4_.H_2_O), 80.00 mg; Mn (MnSO_4_.H_2_O), 60.00 mg; Se (Na_2_SeO_3_), 0.30 mg; I (KI), 0.35 mg. ^2^ The vitamin premix provided the following per kg of diets: vitamin A (trans-retinyl acetate), 9000 IU; vitamin D_3_, 2500 IU; vitamin E (DL-α-tocopherol), 10 IU; vitamin K_3_, 2.65 mg; vitamin B_1_, 2.00 mg; vitamin B_2_, 6.00 mg; vitamin B_6_, 6.00 mg; vitamin B_12_, 0.03 mg; biotin, 0.03 mg; folic acid, 1.25 mg; pantothenic acid, 12.00 mg; and nicotinic acid, 20.00 mg. ^3^ Complex enzymes provided the following enzymes per g: neutral protease, 10,000 U; xylanase, 35,000 U; β-mannanase, 1500 U; β-glucanase, 2000 U; cellulose, 500 U; and amylase, 100 U; pectinase, 10,000 U.

**Table 2 animals-13-03013-t002:** Influence of cysteamine supplementation on egg quality and yolk taurine in hens at the peak stage of egg production (Exp.1).

Cysteamine Supplementation Level	0	0.10%	SEM	*p*-Value
Yolk taurine content (mg/kg)	13.33	19.82	0.424	<0.001
Breaking force (kg/cm^2^)	3.66	3.39	0.093	0.158
Albumen height (mm)	6.66	6.66	0.123	0.984
Haugh unit	85.07	84.57	0.763	0.748
Egg yolk color	5.28	5.80	0.087	0.002

Three yolks in each replicate were mixed as a new yolk sample for taurine content detection. *n* = 6.

**Table 3 animals-13-03013-t003:** Influence of cysteamine supplementation on production performance in hens at the peak stage of egg production (Exp.1).

Item	Feeding Duration (wk)	Cysteamine Supplementation Level (%)		SEM	*p*-Value
		0	0.1		
Laying rate (%)	1	82.06	80.16	0.969	0.329
	2	82.22	80.75	0.965	0.449
	3	83.02	79.70	0.926	0.073
	4	83.02	80.85	0.970	0.266
	5	82.71	82.20	1.216	0.836
Egg mass (g/hen·d)	1	39.34	38.30	0.499	0.303
	2	40.24	39.88	0.512	0.729
	3	41.32	39.55	0.482	0.069
	4	42.24	41.11	0.535	0.291
	5	42.61	42.73	0.653	0.931
Egg weight (g)	1	47.94	47.72	0.134	0.401
	2	48.93	49.32	0.110	0.075
	3	49.78	49.58	0.134	0.446
	4	50.89	50.78	0.158	0.738
	5	51.56	51.97	0.236	0.386
Feed intake (g/d)	1	79.53	77.58	1.144	0.420
	2	83.34	79.71	1.461	0.230
	3	92.71	87.88	1.588	0.133
	4	83.33	85.12	1.652	0.611
	5	92.94	101.05	1.589	0.003
Feed conversion ratio	1	2.02	2.03	0.037	0.703
	2	2.06	2.00	0.031	0.232
	3	2.22	2.24	0.035	0.261
	4	1.99	2.09	0.045	0.217
	5	2.16	2.47	0.094	0.079

**Table 4 animals-13-03013-t004:** Influence of cysteamine supplementation on serum taurine and yolk taurine in hens at the late stage of egg production (Exp.2).

Breed	Cysteamine Supplementation Level (%)	Serum Taurine (mg/kg)	Yolk Taurine Content (mg/kg)
*CAU-3*	0	31.89	17.35
	0.02%	32.24	18.08
	0.04%	29.37	20.43
	0.08%	60.20	22.23
	0.1%	54.46	21.04
Hy-Line brown	0	42.18	17.40
	0.02%	40.78	19.10
	0.04%	47.64	20.39
	0.08%	64.51	22.80
	0.1%	61.98	22.31
SEM		2.359	0.332
Main effect			
Breed	*CAU-3*	41.63 ^b^	19.83
	Hy-Line brown	51.80 ^a^	20.40
Cysteamine level	0	36.01 ^b^	17.38 ^c^
	0.02%	36.12 ^b^	18.59 ^c^
	0.04%	37.67 ^b^	20.41 ^b^
	0.08%	62.16 ^a^	22.52 ^a^
	0.1%	57.87 ^a^	21.68 ^ab^
*p*-value			
Breed		0.005	0.223
Cysteamine level		<0.001	<0.001
Breed × Cysteamine level		0.764	0.871
Linear Quadratic		0.167	<0.001
		0.089	0.051

^a–c^ Values are statistically significant (*p* < 0.05). Three yolks in each replicate were mixed as a new yolk sample for taurine content detection (*n* = 6).

**Table 5 animals-13-03013-t005:** Influence of cysteamine supplementation on production performance in hens at the late laying period (Exp.2).

Item	Feeding Duration (wk)	Cysteamine Supplementation Level (%)					SEM	*p*-Value		
		0	0.02	0.04	0.08	0.1			Linear	Quadratic
Feed intake (g/d)	2	101.24 ^a^	98.54 ^b^	98.18 ^b^	95.04 ^c^	87.62 ^d^	1.021	<0.001	<0.001	0.018
	4	99.04	95.36	96.64	93.11	89.20	1.126	0.052	0.005	0.539
	6	91.12	87.59	90.14	89.44	89.45	1.028	0.885	0.846	0.676
	8	100.64	102.13	100.81	103.89	103.27	0.809	0.663	0.243	0.981
Feed conversion ratio	2	2.09	1.96	2.04	2.06	2.14	0.032	0.535	0.437	0.190
	4	2.07 ^b^	1.92 ^b^	2.01 ^b^	2.15 ^b^	2.52 ^a^	0.047	<0.001	<0.001	<0.001
	6	2.15	2.21	2.07	2.23	2.13	0.035	0.542	0.586	0.653
	8	2.14	2.19	2.30	2.22	2.24	0.025	0.353	0.212	0.225
Laying rate (%)	2	87.16 ^a^	87.56 ^a^	83.75 ^b^	78.51 ^c^	70.07 ^d^	0.535	<0.001	<0.001	<0.001
	4	82.62 ^ab^	87.07 ^a^	82.08 ^b^	73.16 ^c^	60.96 ^d^	0.629	<0.001	<0.001	<0.001
	6	77.14 ^a^	76.45 ^a^	74.35 ^a^	75.11 ^a^	71.22 ^b^	0.629	0.002	0.050	0.220
	8	78.34 ^b^	78.96 ^b^	76.95 ^c^	82.00 ^a^	75.12 ^c^	0.487	0.007	0.320	0.383
Egg mass (g/hen·d)	2	50.06 ^a^	51.05 ^a^	48.97 ^b^	46.44 ^c^	41.85 ^d^	0.312	<0.001	<0.001	<0.001
	4	49.14 ^ab^	50.85 ^a^	48.64 ^b^	43.09 ^c^	36.90 ^d^	0.347	<0.001	<0.001	<0.001
	6	45.26	44.56	43.87	44.01	43.28	0.356	0.315	0.125	0.598
	8	45.88 ^c^	46.04 ^c^	45.557 ^c^	48.94 ^a^	46.75 ^b^	0.272	0.010	<0.001	<0.001
Egg weight (g)	2	59.10 ^b^	58.34 ^c^	58.65 ^c^	59.73 ^a^	60.29 ^a^	0.081	<0.001	<0.001	<0.001
	4	58.80 ^b^	58.70 ^b^	59.07 ^b^	60.18 ^a^	60.41 ^a^	0.080	<0.001	<0.001	0.020
	6	58.94 ^bc^	58.35 ^c^	58.95 ^bc^	59.00 ^b^	60.54 ^a^	0.086	<0.001	<0.001	<0.001
	8	59.03 ^b^	58.46 ^b^	58.83 ^b^	58.56 ^b^	59.97 ^a^	0.078	<0.001	0.056	0.499

^a–d^ Values are statistically significant (*p* < 0.05). *n* = 6.

**Table 6 animals-13-03013-t006:** Influence of cysteamine supplementation on egg quality in hens at the late stage of egg production (Exp.2).

Hen Breed	Cysteamine Supplementation Level (%)	Breaking Force (kg/cm^2^)	Albumen Height (mm)	Haugh Unit	Yolk Color
*CAU-3*	0	3.60	7.87 ^a^	87.49 ^a^	6.33 ^c^
	0.02%	3.25	7.27 ^ab^	84.78 ^ab^	6.78 ^ab^
	0.04%	3.34	6.41 ^bcd^	80.27 ^bcd^	6.40 ^c^
	0.08%	3.26	5.97 ^de^	76.94 ^d^	6.77 ^ab^
	0.1%	2.99	5.87 ^e^	76.48 ^d^	6.58 ^bc^
Hy-Line brown	0	3.59	7.04 ^bc^	82.54 ^bc^	6.92 ^a^
	0.02%	3.57	6.55 ^bcde^	78.91 ^cd^	6.79 ^ab^
	0.04%	3.75	6.60 ^bcd^	79.02 ^cd^	6.80 ^ab^
	0.08%	3.45	6.80 ^bc^	80.98 ^bcd^	6.72 ^ab^
	0.1%	3.48	6.86 ^bc^	81.07 ^bcd^	6.72 ^ab^
SEM		0.039	0.077	0.481	0.030
Main effect					
Breed	*CAU-3*	3.29 ^b^	6.66	81.08	6.57
	Hy-Line brown	3.57 ^a^	6.77	80.50	6.79
Cysteamine level	0	3.59 ^a^	7.42	84.82	6.64
	0.02%	3.55 ^a^	6.90	81.75	6.78
	0.04%	3.42 ^ab^	6.51	79.65	6.60
	0.08%	3.35 ^ab^	6.39	78.96	6.74
	0.1%	3.25 ^b^	6.38	78.84	6.65
*p*-value					
Breed		<0.001	0.462	0.535	<0.001
Cysteamine		0.031	<0.001	<0.001	0.241
Breed × Cysteamine level		0.265	<0.001	<0.001	0.003
Linear		0.003	0.391	0.323	0.309
Quadratic		0.076	0.502	0.776	0.430

^a–e^ Values are statistically significant (*p* < 0.05). *n* = 6.

**Table 7 animals-13-03013-t007:** Influence of cysteamine supplementation on ovary development in hens at the peak stage of egg production (Exp.2).

Breed	Cysteamine Supplementation Level (%)	Relative Ovary Weight (%)	Number of SYF	SYF/Ovary Weight (%)	Number of LYF	LYF/Ovary Weight (%)
*CAU-3*	0	2.98	22.22 ^bc^	4.65 ^c^	7.80 ^ab^	79.57
	0.02%	3.40	20.53 ^bc^	5.31 ^bc^	5.83 ^cd^	86.64
	0.04%	3.12	20.80 ^bc^	7.30 ^ab^	5.60 ^cd^	81.59
	0.08%	3.28	27.80 ^a^	8.13 ^a^	6.44 ^cd^	82.20
	0.1%	2.89	23.52 ^b^	5.72 ^bc^	5.40 ^d^	81.04
Hy-Line brown	0	2.08	19.95 ^c^	4.68 ^c^	6.00 ^cd^	82.25
	0.02%	2.49	22.80 ^bc^	5.44 ^bc^	6.00 ^cd^	82.43
	0.04%	2.59	16.27 ^d^	3.96 ^c^	6.83 ^bc^	84.08
	0.08%	2.35	20.18 ^bc^	5.37 ^bc^	6.20 ^cd^	81.73
	0.1%	2.64	21.96 ^bc^	4.57 ^c^	8.33 ^a^	81.39
SEM		0.078	0.477	0.248	0.168	0.497
Main effect						
Breed	*CAU-3*	3.13 ^a^	22.97	6.22	6.21	82.21
	Hy-Line brown	2.43 ^b^	20.23	4.80	6.67	82.38
Cysteamine level	0	2.53	21.08	4.67	6.90	80.91
	0.02%	2.94	21.66	5.37	5.92	84.54
	0.04%	2.85	18.54	5.63	6.22	82.84
	0.08%	2.82	23.99	6.75	6.32	81.97
	0.1%	2.77	22.74	5.14	6.87	81.22
*p*-value						
Breed		<0.001	<0.001	0.001	0.081	0.861
Cysteamine level		0.318	<0.001	0.035	0.092	0.128
Breed × Cysteamine level		0.350	<0.001	0.031	<0.001	0.157
Linear		0.540	0.155	0.931	0.423	0.798
Quadratic		0.277	0.357	0.314	0.751	0.287

^a–d^ Values are statistically significant (*p* < 0.05). SYF, small yellow follicles (4 mm ≤ diameter < 8 mm); LYF, large yellow follicles (diameter ≥ 8 mm); relative ovary weight means the ratio of ovary weight to body weight. *n* = 6.

## Data Availability

The data presented in this study are available on request from the corresponding author.

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
