# Peer review of "Impact of Dietary Supplementation of Cysteamine on Egg Taurine Deposition, Egg Quality, Production Performance and Ovary Development in Laying Hens"

_animals, 2023, doi:10.3390/ani13193013_

Round 1
Reviewer 1 Report
1. Make sure you use the same font size and type all over the document (font size in simple summary is smaller than the rest)
2. Improve your simple summary
3. Check for grammar in line 58 (ecommon), line 63 benefit for kidneys... no renal,
4. Please make sure you use the same font size
5. Please change Mcal to Kcal/kg to be more exact
6. Explain how did you measure egg color?
7. Please improve result tables to show data clearer
8. Please be sure to clearly label what tables belong to experiment 1 and experiment 2
Authors need to correct some grammar errors
Author Response
- Make sure you use the same font size and type all over the document (font size in simple summary is smaller than the rest)
Response: We have checked all of the manuscript and revised the font size.
- Improve your simple summary.
Response: The simple summary has been revised as “Taurine is necessary amino acid for human health, while cysteamine is an intermediate metabolite to the synthesis of taurine. So, this study investigated effects of dietary cysteamine supplementation on the egg taurine deposition efficiency, egg quality, production performance and ovary development in laying hens. The results of this study indicate that cysteamine supplementation benefits yolk taurine deposition in both peak and late stage of egg production, but hens at the late stage of egg production show a depressed production performance and egg quality. The present study reveals that laying hens at the peak stage of egg production are suitable for cysteamine diets to produce high taurine eggs.”
- Check for grammar in line 58 (ecommon), line 63 benefit for kidneys... no renal,
Response: “ecommon dietary nutrients” has been revised as “fundamental dietary nutrients”. “renal” has been revised as “kidneys”.
- Please make sure you use the same font size
Response: We have checked all of the manuscript and revised the font size.
- Please change Mcal to Kcal/kg to be more exact.
Response: We have revised “Mcal” to “Kcal” and revised the corresponding metabolizable energy number.
- Explain how did you measure egg color?
Response: Eggs were collected and immediately examined by the Model EMT-7300, a machine made in Japan that can measure egg quality such as egg weight, egg color, egg breaking force and so on. According to the instruction book, the color measured by this machine is based on robotmation yolk color chart ranged from 1.0 – 15.0m. (https://www.damarus.com/product/egg-multi-tester-emt-7300/).
- Please improve result tables to show data clearer
Response: The tables display weekly data information, resulting in each indicator not being very clear at first glance. However, we also looked at other literature and found that everyone was using a three-line table, and we really did not know how to change the table.
- Please be sure to clearly label what tables belong to experiment 1 and experiment 2
Response: Exp.1 and Exp.2 has been labeled in each table.
Reviewer 2 Report
The paper under consideration investigates the impact of dietary cysteamine supplementation on yolk taurine content in hens during various stages of egg production. The study is divided into two experiments, focusing on hens at different production stages and varying cysteamine supplementation levels. This review will evaluate the key findings, methodology, and implications of the research. The study employs a well-structured experimental design involving two distinct experiments. In the first experiment, CAU-3 hens at the peak of egg production were used to examine the effects of a 0.1% cysteamine-supplemented diet on yolk taurine content, egg quality, and production performance. The second experiment extended the investigation to two different breeds of hens at the late stage of egg production, assessing the influence of cysteamine supplementation levels ranging from 0% to 0.10% on various parameters, including yolk taurine content, egg quality, production performance, and ovary development. Cysteamine supplementation at 0.1% significantly increased yolk taurine content without adversely affecting production performance or egg quality. This finding suggests that cysteamine can positively impact taurine deposition in yolks during peak egg production. The highest yolk taurine content was observed with cysteamine supplementation at 0.08%, indicating a dose-dependent effect on taurine deposition in yolks. Notably, cysteamine supplementation led to a temporary reduction in production performance during the initial weeks of feeding, but this negative impact diminished with continued feeding.
The paper has several notable shortcomings that need to be addressed:
- Inadequate Title and Simple Summary: The title of the paper should accurately reflect the content of the study. However, it seems that the paper's summary does not align with the title, indicating a lack of clarity and focus. This can mislead readers and make it challenging to understand the paper's main objectives and findings. The summary should be more comprehensive and informative.
- Statistical Analysis Issues: The paper appears to suffer from statistical analysis problems. It mentions the use of ANOVA and interaction tests without proper explanations. Authors should provide clear explanations of both statistical methods, including when and why they were used in the study. Additionally, the paper should detail the specific statistical tests applied, including assumptions made and results obtained, to ensure transparency and replicability.
- Lack of Practical Implications: The paper fails to discuss the practical implications of its findings regarding egg yolk taurine. Authors should address whether the deposition of taurine in egg yolks is beneficial or detrimental for both laying hens and consumers. Providing this information is crucial for understanding the significance and real-world applications of the research.
- Unclear Rationale for Breed and Age Selection: The paper does not adequately justify the choice of two different breeds at different ages. It should provide a clear rationale for why these specific conditions were selected, as this choice impacts the generalizability of the results. Without a valid justification, it raises questions about the study's design and methodology.
- Ad Libitum Ration Feeding: The paper mentions that laying hens were given feed ad libitum, which is a significant departure from the standard practice of feeding laying hens with calculated rations. Authors should explain the reasoning behind this choice and discuss how it might have influenced the results. Without this explanation, it's challenging to assess the validity of the study's conclusions.
In summary, the paper requires substantial improvements in terms of clarity, statistical analysis transparency, practical implications, and justification for its design choices. Addressing these issues would enhance the quality and relevance of the research.
Author Response
The paper has several notable shortcomings that need to be addressed:
- Inadequate Title and Simple Summary: The title of the paper should accurately reflect the content of the study. However, it seems that the paper's summary does not align with the title, indicating a lack of clarity and focus. This can mislead readers and make it challenging to understand the paper's main objectives and findings. The summary should be more comprehensive and informative.
Response: According to your suggestion, we revised the title and simple summary. The title “Dietary supplementation of cysteamine promotes taurine deposition in eggs” has been revised as “Impact of dietary supplementation of cysteamine on egg taurine deposition, egg quality, production performance and ovary development in laying hens”.
The simple summary has been revised as “Taurine is necessary amino acid for human health, while cysteamine is an intermediate metabolite to the synthesis of taurine. So, this study investigated effects of dietary cysteamine supplementation on the egg taurine deposition efficiency, egg quality, production performance and ovary development in laying hens. The results of this study indicate that cysteamine supplementation benefits yolk taurine deposition in both peak and late stage of egg production, but hens at the late stage of egg production show a depressed production performance and egg quality. The present study reveals that laying hens at the peak stage of egg production are suitable for cysteamine diets to produce high taurine eggs.”
- Statistical Analysis Issues: The paper appears to suffer from statistical analysis problems. It mentions the use of ANOVA and interaction tests without proper explanations. Authors should provide clear explanations of both statistical methods, including when and why they were used in the study. Additionally, the paper should detail the specific statistical tests applied, including assumptions made and results obtained, to ensure transparency and replicability.
Response: Thank you for your suggestions. We now revised it as “In Exp. 2, a 2×5 factorial arrangement of 2 breeds and 5 dietary cysteamine levels was used, and the main effect of breed, dietary cysteamine level, and interactions between breed and dietary cysteamine level were tested by general linear model. Items with significant interaction effects were further analyzed by ANOVA.”
- Lack of Practical Implications: The paper fails to discuss the practical implications of its findings regarding egg yolk taurine. Authors should address whether the deposition of taurine in egg yolks is beneficial or detrimental for both laying hens and consumers. Providing this information is crucial for understanding the significance and real-world applications of the research.
Response: Yes, we ignored this key question. Now, we have discuss it as follows: Taurine can be synthesized by catalysis of cysteinesulfinate decarboxylase. It has been shown that taurine exists in many different tissues in different organisms [1]. In this study, we found that yolk taurine content was enhanced with 0.1% cysteamine sup-plementation in hens at the peak stage of egg production or at more than 0.04% in hens at the late stage of egg production. This indicates that cysteamine can be used to syn-thesize taurine in hens at multiple stages of production. Though it has been reported that serum taurine content declined with age in mice, monkeys, and humans [7], we found that hens at the late stage of egg production (aged 68 wk) produce more taurine than hens at the peak stage of egg production when fed basic diets. This suggests hens at the late stage of egg production may have a greater ability to synthesize taurine relative to younger birds. However, with diets supplemented with cysteamine, it seems that the effect induced by the stage of egg production is lost. This may imply that the basic diet for hens at the peak stage of egg production is more deficient in raw materials for tau-rine synthesis than diets for hens at the late stage of egg production.
- Unclear Rationale for Breed and Age Selection: The paper does not adequately justify the choice of two different breeds at different ages. It should provide a clear rationale for why these specific conditions were selected, as this choice impacts the generalizability of the results. Without a valid justification, it raises questions about the study's design and methodology.
Response: Thank you for your constructive suggestion. It's true that our experimental design is not perfect. The reasons are as follows: Since the efficacy of cysteamine was not clear, so Exp.1 was conducted was used to explore if cysteamine supplementation affect egg taurine deposition. We found that cysteamine supplementation benefit for egg taurine deposition in Exp.1 and wanted to do further study. Since the hens in this study were bought from a commercial farm, hens of this flock were go into the late stage of egg production. Besides, COVID-19 pandemic made it difficult to transport, which made us couldn’t find hens at the peak stage of egg production. Considering that different breeds of laying hens at different stages of egg production have different methionine requirements and different ability to metabolize cysteamine, the study was conducted on the hens at the late stage of egg production as test animals, so that the effect of cysteamine supplementation on hens at the late stage of egg production could be observed.
- Ad Libitum Ration Feeding: The paper mentions that laying hens were given feed ad libitum, which is a significant departure from the standard practice of feeding laying hens with calculated rations. Authors should explain the reasoning behind this choice and discuss how it might have influenced the results. Without this explanation, it's challenging to assess the validity of the study's conclusions.
Response: Thank you for your question. We discussed and recalled the procedure of the experiment and found that this is a mistake written. We have revised it as “Feed was provided at 7:00 am and 1:00 pm, and water was pro-vided ad libitum. The feeding amount each day was appropriately adjusted according to the amount of feed remaining from the previous day. This was done to ensure no feed was leftover in the trough each night and prevent picky eating.”
Reviewer 3 Report
In this interesting manuscript, the authors evaluated the effect of dietary cysteamine supplementation on the deposition of taurine in eggs. Taurine is an essential free amino acid for the body. Consumption of foods rich in taurine is beneficial to people's health. Cysteamine has been found to reduce the steps of taurine synthesis. The authors found experimentally that cysteamine supplementation during peak laying can improve yolk taurine content without affecting egg quality and production performance. However, supplementation of the diet with cysteamine during the late stage of laying can improve the taurine content in the egg yolk, while reducing production performance and egg quality. With editing and some revisions, I feel that this manuscript will be suitable for publication.
1.Line 19:“Taurine is a necessary amino acid for human health.”There's a grammatical error in this sentence. Please revise it.
2.Why were only China Agricultural University-3 hens chosen as the experimental animal for Experiment 1 while Hyland Brown and China Agricultural University-3 hens were chosen for Experiment 2?
3.How to determine the range of dietary additions of cysteamine?
4. Why was only the effect of adding 0.1% cysteamine to the diet at the peak stage of egg production on yolk taurine content, egg quality and performance tested?
5. Line 176-179:Please keep the format consistent with the preceding text.
6. Line 189: Change the value in rpm to g.
7. What are the criteria for evaluating Egg yolk color?
8. Explain briefly the methods of Albumen height and Haugh unit determination.
9. Please provide a brief description of the relationship between Breaking force, albumin height, Haugh unit, Egg yolk color and egg quality.
Extensive editing of English language required
Author Response
1.Line 19: “Taurine is necessary amino acid for human health”. There's a grammatical error in this sentence. Please revise it.
Response: This sentence has been revised as “Taurine is a necessary amino acid for human health”.
2.Why were only China Agricultural University-3 hens chosen as the experimental animal for Experiment 1 while Hyland Brown and China Agricultural University-3 hens were chosen for Experiment 2?
Response: In the Exp.1, we used CAU-3 hens at the peak stage of egg production and found that cysteamine supplementation (0 and 0.1%) is benefit for egg taurine deposition. Then we wanted to know how cysteamine affects hens at the late stage of egg production in Exp.2. However, we used two breeds of hens (CAU-3 hens and Hyline Brown hens) in Exp.2, and then compared the effects of cysteamine on the two breeds of hens, make it unclear rationale for breed and age selection. The Best design may be both breeds of hens were used in both Exp.1 and Exp.2. Even though the imperfect design, this study provides us a lot of information about the use of cysteamine in laying hens.
3.How to determine the range of dietary additions of cysteamine?
Response: Our dosage is mainly based on published literature (doi:10.3382/ps.2008-00040) by Hu et al., (2008). Previous study has study the effect of cysteamine supplementation at 0.04% in laying hens. Our study showed that cysteamine supplementation at 0.1% still benefit egg taurine deposition and without negative influence on egg quality and production performance in hens at the peak stage of egg production. And different dosage of cysteamine was also studied in hens at the late stage of egg production.
- Why was only the effect of adding 0.1% cysteamine to the diet at the peak stage of egg production on yolk taurine content, egg quality and performance tested?
Response: The Exp.1 was just used to explore if cysteamine supplementation promote egg taurine deposition. So, we didn’t do so many groups.
- Line 176-179: Please keep the format consistent with the preceding text.
Response: Thank you for your carefully reading, we have revised all of the punctuations. For example, “vitamin A (trans-retinyl acetate) 9000 IU, vitamin D3 2500 IU,” was revised as “vitamin A (trans-retinyl acetate), 9000 IU; vitamin D3, 2500 IU;”.
- Line 189: Change the value in rpm to g.
Response: 10000 rpm has been revised as 10600 g
- What are the criteria for evaluating Egg yolk color?
- Explain briefly the methods of Albumen height and Haugh unit determination.
Response for Q7 and Q8: Eggs were collected and immediately examined by the Model EMT-7300, a machine made in Japan that can measure egg quality such as egg weight, egg color, egg breaking force and so on. According to the instruction book, the egg yolk color, albumen height and Haugh unit were measured (https://www.damarus.com/product/egg-multi-tester-emt-7300/).
- Please provide a brief description of the relationship between Breaking force, albumin height, Haugh unit, Egg yolk color and egg quality.
Response: Relationship of these index and egg quality has been added in discussion.
Such as, “Egg quality includes the index of yolk color, albumin height, Haugh unit, breaking force and so on. Usually, the higher the yolk color of an egg, the more popular it is with consumers, and yolk color is influenced by pigment content in feed, hen breed, and antioxidant contents [23]”. Line 391-394
“Albumin height reflects the protein quality and is influenced by storage time and age of hens. Silversides et al. (2004) reported that albumin height is decreased with age and storage time”. Line 405-408
“High value of breaking force means the hard shell and the less likely the egg will break. Line 411-412
An egg with Haugh unit ≥ 80 means it’s a fresh egg, because Haugh units will decrease as the storage time is extended [29]”. line 416-417
Round 2
Reviewer 2 Report
accepted